psychology

temporal discounting, perceived relative affluence, social trust, information shock

**Authors for correspondence:**
Léonard Guillou
e-mail: leonard.guillou@gmail.com
Coralie Chevallier
e-mail: coralie.chevallier@gmail.com

# Temporal discounting mediates the relationship between socio-economic status and social trust

Léonard Guillou[1], Aurore Grandin[2] and Coralie Chevallier[2]

[1]Institut Jean Nicod, Département d'études cognitives, ENS, EHESS, CNRS, PSL Research University, 75005 Paris, France
[2]Laboratoire de neurosciences cognitives et computationnelles, Département d'études cognitives, École normale supérieure, INSERM U960 Paris, France

 LG, 0000-0003-2623-9493; AG, 0000-0002-3666-9270; CC, 0000-0002-7358-4962

Social trust and income are associated both within and across countries, such that higher income typically correlates with increased trust. While this correlation is well-documented, the psychological mechanisms sustaining this relationship remain poorly understood. One plausible candidate is people's temporal discounting: on the one hand, trust has a strong time component—it exposes the individual to immediate costs in exchange of uncertain and delayed benefits; on the other hand, temporal discounting is robustly influenced by income. The goal of our studies was to test whether temporal discounting mediates the relationship between income and trust and whether experimentally manipulating perceived income has a downstream impact on temporal discounting and trust. To do so, participants who underestimated their relative income position received information about their true position in the income distribution in order to correct their misperception. Our results indicate that temporal discounting partially mediates the effect of income on social trust in a pre-registered online study on British participants ($N = 855$). However, receiving a positive information shock on one's income position had no impact on either temporal discounting or social trust. In a second pre-registered study, we replicated the finding that temporal discounting partially mediates the effect of income on social trust in a representative sample of the British population ($N = 1130$).

## 1. Introduction

Social trust can be defined as the belief that other people are reliable cooperation partners. When one decides to take part in

a cooperative interaction, it makes sense to invest in the interaction to the extent that one can trust that the cooperation partner will provide an adequate return on investment. This return can take the form of later reciprocation or of a reputational gain. Formally then, social trust is the belief that most people will reciprocate—offer a return on investment—if one invests in a social interaction with them.

Social trust is a belief central to decision-making processes and it affects a range of prosocial behaviours like cooperation in situations of conflict of interests [1], acceptance of vaccination [2], or cooperation in social dilemmas with a large number of actors [3]. At the societal level, social trust is linked to a host of positive outcomes, such as improved economic growth [4,5]. But despite its positive outcomes, social trust varies considerably between individuals and societies [6]. Why is this the case? Do variations in social trust follow a predictable pattern? Can these variations be traced back to specific psychological and environmental features?

One factor explaining the variability of social trust is that it is sensitive to socio-economic status. Existing works suggest that childhood socio-economic circumstances [7,8] as well as current income [9] are positively associated with social trust. Brandt *et al.* [10] also showed with longitudinal data, that increases in socio-economic status predict increases in social trust, and that this model is more viable than a model using increases in social trust to predict increases in socio-economic status. Similarly, exogenous negative shocks on childhood resources have an impact on adult levels of social trust. Using data from Germany after the Second World War, Hörl *et al.* [11] showed that children exposed to more hunger during childhood had lower levels of trust as adults. Petersen & Aarøe [12] even found an effect of lower birth weight on social trust in adulthood.

While the association between social trust and socio-economic status is robustly established, the psychological mechanism by which this takes place is not known. An interesting characteristic of social trust is its temporal component, underlined by researchers from several fields, such as education [13], sociology [14] or psychology and neuroscience [15], who agree on the future-oriented character of social trust. For instance, Krueger & Meyer-Lindeberg [15] define social trust as accepting one's vulnerability towards one's expectation regarding 'the behaviour of another party that will produce some positive outcome in the future'. Social trust is based on reciprocity [15] and, thus, is associated with certain and immediate costs and with uncertain and delayed benefits.

Therefore, one possibility is that social trust reacts to individual variations in temporal preferences, which are themselves affected by socio-economic status. People with short-term preferences should therefore be less prone to investing in social interactions, for which the return on investment is delayed. In terms of behaviours, the trade-off would be between choosing the sooner smaller reward of defecting over the later larger reward offered by reciprocation from the interaction partner or a reputational gain. In a recent paper, Pepper & Nettle [16] suggested that the varying length of time-horizon between individuals could explain several socio-economic gradients in social behaviours, including financial decisions [17,18], educational attainment [17], academic engagement and performance [19], smoking [20] or eating behaviours [21].

In this paper, we apply this logic to social trust: our first hypothesis is that temporal discounting is a key psychological mechanism by which socio-economic status impacts social trust. In other words, we hypothesize that temporal discounting mediates the relationship between socio-economic status and social trust.

The link between temporal discounting and social trust has only been tested in a couple of studies, with mixed findings. A first study, conducted in Vietnam, did not show any effect of temporal discounting on trust, using a trust game [22], but this null effect might be due to a lack of statistical power ($N = 156$ only). A second study found that temporal discounting is an important covariate of self-reported trust [23] but in this study, temporal discounting was measured with a single question, which limits the accuracy of the temporal discounting estimate. Our experimental design tackles both limitations and allows us to test whether variations in temporal discounting are associated with variations in social trust inside a mediation model. To avoid the pitfalls of the two studies above-mentioned, we used well-validated measures of time discounting and social trust, and we made sure to have sufficient statistical power.

Moving beyond correlations, the goal of our paper is also to study the causal structure of this association by experimentally manipulating perceived income. Income [24–26] and relative deprivation [27,28] are indeed well-known drivers of temporal preferences. Economists typically measure temporal preferences using delay discounting experiments in which participants are asked to choose between a smaller but sooner reward—for example $2 tomorrow—and a larger later reward—for example, $5 in one week. In these types of experiments, lower socio-economic status individuals consistently display steeper temporal discounting [24–26]. Less-educated adolescents, compared to adolescents with higher

education, also show the same pattern [29]. Callan *et al.* [27] even found that an experimental manipulation of personal relative deprivation increased participants' impatience in a temporal discounting task: those who learnt that they had less income than similar others displayed higher temporal discount rates. In this context, our second hypothesis is that inducing fluctuations in temporal discounting by manipulating perceived relative income should impact social trust.

To manipulate socio-economic status, we capitalize on work demonstrating that many people underestimate their wealth and that it is possible to correct these misperceptions, with measurable effects on attitudes and behaviours. For instance, faulty beliefs about inequalities are common in the US [30] or in Australia [31]. In both countries, people underestimate the level of wealth inequality in their country. The same occurs for one's own position in the income distribution, for example, in Sweden [32] or in Argentina [33]. In these countries, people underestimate their relative position in their country's income distribution. These misperceptions can be explained by people's over-reliance on cues from their local environment [34]. Interestingly, studies have shown that providing economic information (for instance, about inequalities or income distribution) as an experimental treatment can have an impact on people's attitudes, e.g. their preference for redistribution [32,33] or tolerance for inequalities [35]. In this paper, we apply a similar method in order to test whether experimentally manipulating perceived income has a downstream impact on temporal discounting and trust.

To test our hypotheses, we designed two online studies. In study 1, we tested a causal mediation model with time discounting mediating the effect of socio-economic status on social trust. To do so, we used an experimental design by randomly informing a subsample of participants about their true relative income position in the British society. Since objective socio-economic status cannot be easily manipulated experimentally (except by giving people extra income), we focused on the way people perceive their resource level (i.e. their perceived relative income). Specifically, we corrected people's perception of their income when they felt that their relative income position was lower than it actually was. In other words, the goal of our intervention was to provide a 'positive psychological income shock'. Our main question was whether correcting people's misperception of their relative income would influence temporal discounting and as a consequence, social trust. We hypothesized that respondents who learnt that they had a higher relative income than they thought (positive income shock) would discount future benefits less, and that as a result they would display more social trust. The experiment targeted only participants who hold wrongly negative beliefs about their position in society. For ethical reasons, participants who believed they had a higher relative income than they actually had, did not receive any correction. We also studied this mediation pathway with a correlational approach testing if the effect of being of low or high socio-economic status on social trust was mediated by temporal discounting.

In study 2, we replicated our results from study 1, particularly the mediating role of temporal discounting on the relationship between socio-economic status and social trust and between time discounting and social trust. In this study, we did not test any treatment, data were only observational. This replication was made on a sample representative of the British population.

## 2. Study 1

In this study, we tested whether temporal discounting mediates the relationship between income and trust and we tested the causal influence of perceived relative income on people's temporal discounting and social trust. To do so, we used an experimental design by randomly informing a subsample of participants about their true relative income position in the British society. Our main question was whether this income shock about participants' actual relative income could influence temporal discounting and as a consequence, social trust. We hypothesized that respondents who learnt that they were underestimating their relative income (and who thus benefited from a positive income shock) would discount future benefits less, and that as a result they would display more social trust. The experiment targeted only participants who wrongly held negative beliefs about their relative income. For ethical reasons, participants who overestimated their relative income did not receive any correction.

### 2.1. Methods

#### 2.1.1. Participants

Participants were recruited online via Prolific Academic (https://www.prolific.co). We used Prolific's pre-screening criteria to filter participants by nationality and approval rate. We recruited only British

participants with a minimum approval rate of 90%. Participants were excluded if they reported a personal monthly income above £12 500, which corresponds to the highest tax bracket and top-earning percentile in the UK, in order to minimize potential income reporting mistakes. We also excluded students because their reported income may not reflect their actual living standards (e.g. students may report no income but receive parental support).

To determine sample size, we set an *a priori* power level and a minimum detectable effect that the intervention was expected to have on temporal discounting [36]. We used the formula recommended by Bloom [37]: $MDE = (t_{1-k} + t_{\alpha/2}) * \sqrt{1/P(1-P)} * \sqrt{\sigma^2/N}$, where MDE is minimum detectable effect; $N$ is number of observations; $\sigma$ is standard deviation of effect; $P$ is proportion of treated (50%); $t_{\alpha/2} = 1.96$ (using a two-tailed hypothesis at the 0.05 significance level); $t_{1-k} = 0.84$ (setting statistical power to 80%). We based our power calculation on measures of temporal discounting taken in a British sample ([38], see pre-registration: https://osf.io/vq8t3). We found that we needed 807 participants to identify a standardized minimum detectable effect of 0.2. To compensate for attrition, we recruited 888 participants, which corresponds to 807 plus 10% attrition. The study used for the power analysis ([38], see pre-registration: https://osf.io/vq8t3) did not contain any question on social trust. Therefore, we had no means of knowing whether a study with 888 participants was sufficiently powered to detect an effect of an income shock on social trust mediated by temporal discounting.

### 2.1.2. Materials

The questionnaires and discounting task were presented using Qualtrics (https://www.qualtrics.com).

*Socio-economic status.* We included two measures of socio-economic status: an objective measure and a subjective measure. Objective socio-economic status refers to income and education. We asked respondents about their personal income rather than their household overall income, because studies have shown that questions about household income induce lower response rates and produce lower quality data [39]. Participants were asked to report their personal monthly income in a free-text box in order to avoid unintentional priming effects that come with the use of income brackets [40]. Moreover, in order to minimize reporting mistakes such as reporting yearly instead of monthly income, or unintentionally typing an additional zero, we calculated their annual income based on the monthly income they reported and asked them to confirm that it corresponded to their actual earnings. In addition to personal income, we asked participants to specify which amount of benefits they received (jobseeker's allowance, incapacity benefits, etc.).

We acknowledge that it is not only income or occupation that generate differences in behaviour but the experience of various deprivations that are often associated with being of lower socio-economic status [16]. Therefore, we also computed a measure of perceived relative affluence (subjective socio-economic status) combining the MacArthur scale of subjective social status [41] and Griskevicius *et al.*'s [42] three-item questionnaire.

We used a slightly modified version of the MacArthur scale in order to focus on income: 'Think of this scale as representing where people stand in the UK. At the top of the scale (10) are the people who are the best off in terms of overall income. At the bottom (1) are the people who are the worst off in terms of overall income. Where would you place yourself in this scale?' Griskevicius *et al.*'s [42] three-item questionnaire was: 'How much do you agree with each of the below statements on a scale from 1 (strongly disagree) to 7 (strongly agree)? My family and I have had enough money for things. I have lived in a relatively wealthy neighbourhood. I have felt relatively wealthy compared to other people in my neighbourhood.'

*Temporal discounting.* Prior research has demonstrated that hypothetical delay-discounting tasks reliably predict actual temporal preferences [43], and correlate with real-world measures of impulsivity such as smoking, overeating, and credit card debt [26,44,45]. Our temporal discounting task was based on the one designed by Frye *et al.* [46]. In this task, participants had to complete three blocks of an intertemporal choice task with varying delays (three weeks, three months and 2 years) and varying amounts. All values displayed to participants were rounded to the nearest 50 pence (£0.50). Each block consisted of six binary choice trials. The task ended with two catch trials (e.g. Do you prefer £30 in 3 days or £1 in 2 years?) to make sure that participants read carefully, resulting in a total of 20 trials. In the first block, participants had the choice between a smaller reward in 3 days, and a larger reward in three weeks. In the second and third blocks, the later delay was set to three months and 2 years, respectively. The choices were hypothetical, which means that participants did not actually receive the money corresponding to their choices. As in Haushofer *et al.*'s paper [47], possible serial correlations and order effects in participants' responses were controlled for by randomizing the order of trials across blocks. In order to control for

possible unintended effects, the position of the sooner-smaller and later-larger alternatives on the screen (top versus bottom) was also randomized across trials. The later reward was kept constant at £70 (as in [26]), while the smaller reward was adjusted according to each participant's choices. The adjustment was based on a bisection algorithm, following [46]. As recommended by the authors, the adjustment for the upcoming trial was always equal to the maximum amount multiplied by $2^{-n}$, where $n$ was the trial number for the current adjustment.

Such a procedure allowed us to identify three individual indifference points for each participant, which can be used as the basis for the computation of discount rates. In our case, the indifference point for a particular delay is the amount of money to which a participant is indifferent between receiving it in three days and receiving £70 with the particular delay. Given that we only have three indifference points per participant, we were not able to fit curvilinear regression models (like Mazur's [48], Green et al.'s [24] or Rachlin's [49]) and focused instead on the area under the curve (AUC) [46]. This measure carries partial information, because different curve shapes can result in the same area under the curve. We focused on AUC because our main goal was to estimate between-participants differences in temporal discounting steepness rather than to precisely model participants' discounting curves. Using area under the curve was therefore an optimal choice considering the trade-off between measurement precision and study duration.

*Social trust*. Social trust is a complex behaviour, which is difficult to fully capture in a brief study, but studies suggest that questionnaires are often better than economic games in predicting real-life behaviours [50–52]. Here, we focused on three questions that are routinely used in national and international surveys, such as the European Values Study: 'Generally speaking, would you say that most people can be trusted or that you can't be too careful in dealing with people?', 'Would you say that people usually only take care of themselves or that they try to be helpful most of the time?' and 'Do you think that most people would try to take advantage of you if they had the opportunity or that they would try to be fair?'. Answers available to participants for the first question were 'most people can be trusted', 'can't be too careful' or 'don't know'. A 1–10 scale was used for the second and third questions.

### 2.1.3. Procedure

Participants started with questions about socio-economic status and demographics. We used these answers to calculate how much our participants underestimated or overestimated their relative position in the British income distribution. We calculated the sum of their reported income and taxable benefits to determine which decile they belonged to. We used statistics from the British government's Personal Income statistics release (https://www.gov.uk/government/collections/personal-incomes-statistics) as a reference to compute participants' position in the income distribution, which provides percentiles of total annual income (comprising taxable benefits). We compared this decile to their answer to the MacArthur scale, so as to estimate how much participants had a biased perception of their position in the income distribution. We defined the bias of a participant as the difference between her perceived and actual income decile. Participants who underestimated (/overestimated) their relative income by one decile point or more were categorized as having a negative (/positive) bias. The remaining participants were defined as having no bias. The survey stopped there for participants who estimated their position accurately or overestimated it (null or positive bias). Those who underestimated their position (negative bias) were randomly assigned to the experimental or control condition. In the experimental condition, participants were presented with a correction of their misperception of relative income, based on their reported income and taxable benefits. These participants therefore received a positive income shock. Participants then completed the temporal discounting task. Next, they answered questions on environmental attitudes for another study. Questions on social trust came last.

## 2.2. Data treatment

All data treatment and analyses were carried out in R. They were all pre-registered (see: https://osf.io/pyj24). The only deviation from the pre-registration plan is that the exclusion for reaction times was applied at the single-trial level. In what follows, we focus on the mediation analyses.

*Exclusion of participants*. Participants were excluded from all analyses if they failed one or more catch trials in the temporal discounting task. They were also excluded if they responded too fast (less than 500 ms) or too slowly (greater than 2 min) to single question screens or if they responded too fast

(less than 3 s) or too slowly (greater than 5 min) to the other survey pages (with more than one question per screen). We used these filters to make sure that participants were attentive and read the questions carefully. This left a sample of 855 participants (427 participants in the control group and 428 in the treated group), with 442 women. Ages ranged 21 to 72, $M = 39.76$, s.d. = 10.46.

*Socio-economic status*. We computed total income, which was the sum of participants' personal monthly income and benefits. Education level was transformed from an unordered categorical variable (e.g. Completed College) to a 1–6 score. Objective socio-economic status was the sum of $z$-scored total income and $z$-scored education level. Subjective socio-economic status was the sum of $z$-scored items Griskevicius *et al.*'s [42] questionnaire and MacArthur scale.

*Information shock* (treatment) corresponds to a binary variable indicating whether a participant was in the control group or the treated group.

*Temporal discounting*. To calculate individual temporal discount rates, we isolated the indifference point for each delay and for each participant and then calculated the area under the discounting curve linking these points [53]. The AUC between two points on the curve is calculated as $(x_2 - x_1)[(y_1 + y_2)/2]$, where $x_1$ and $x_2$ are the successive delays and $y_1$ and $y_2$ are the indifference points for those delays. Then the AUC between three weeks and three months and the one between three months and 2 years were summed, resulting in a single value of AUC per participant. Lower values of AUC indicate steeper discounting [46].

*Social trust*. A general trust binary variable was constructed from the first question ('Generally speaking, would you say that most people can be trusted or that you can't be too careful in dealing with people?'), with a value of 1 for individuals responding 'most people can be trusted' and a value of 0 for the others. A 1–10 scale was used for the two additional questions ('Would you say that people usually only take care of themselves or that they try to be helpful most of the time?' and 'Do you think that most people would try to take advantage of you if they had the opportunity or that they would try to be fair?'). The answers obtained to these three questions were $z$-scored and summed to obtain a single index capturing participants' beliefs on others' level of prosociality.

## 2.3. Statistical analyses

*Correlational mediation model*. Our first main hypothesis (H1) is that the relationship between social trust and socio-economic status is mediated by individual discount rates. To test the mediation model, we used the causal-steps procedure described by Baron & Kenny [54]. Following recommendations from Miller *et al.* [55] and Aguinis *et al.* [56], we tested the significance of the indirect pathway with the R mediation package [57] using the function mediate. Despite the name of this package: 'R package for causal mediation analysis', the mediation analysis described in this section is not causal because it is not based on a treatment but only on observational data. In the present case, the treatment variable (i.e. the 'treat' argument in the mediate function) is socio-economic status (alternatively objective or subjective).

*Causal mediation model*. Our second main hypothesis (H2) is that the effect of the positive information shock on social trust is mediated by individual discount rates. To test this hypothesis, we used the same steps as for H1 but with the binary variable 'treatment' (0: control, 1: treated) as the treatment variable (i.e. the 'treat' argument in the mediate function).

## 2.4. Results

For descriptive statistics, see table 1.

For a graph of temporal discounting curves, see figure 1.

*Correlational mediation model*. To test our first main hypothesis (H1), we conducted regression analyses relying on the causal-steps procedure set forth by Baron & Kenny [54] to detect a mediation effect. The results suggest a partial mediation: the effect size of both objective and subjective (table 2) socio-economic status is lowered by the addition of temporal discounting in the model. For both mediation pathways (objective versus subjective socio-economic status), the positive effect of socio-economic status on social trust is partially mediated by temporal discounting (objective socio-economic status: average causal mediating effect of temporal discounting: $\beta = 0.02$, $p = 0.001$, proportion of the total effect that is mediated: 9.0%, $p = 0.001$; subjective socio-economic status: average causal mediating effect of temporal discounting: $\beta = 0.02$, $p = 0.01$, proportion of the total effect that is mediated: 6.2%, $p = 0.01$). These values are calculated for control.value = −0.6 (first quartile) and treat.value = 0.5 (third quartile) for objective socio-economic status and

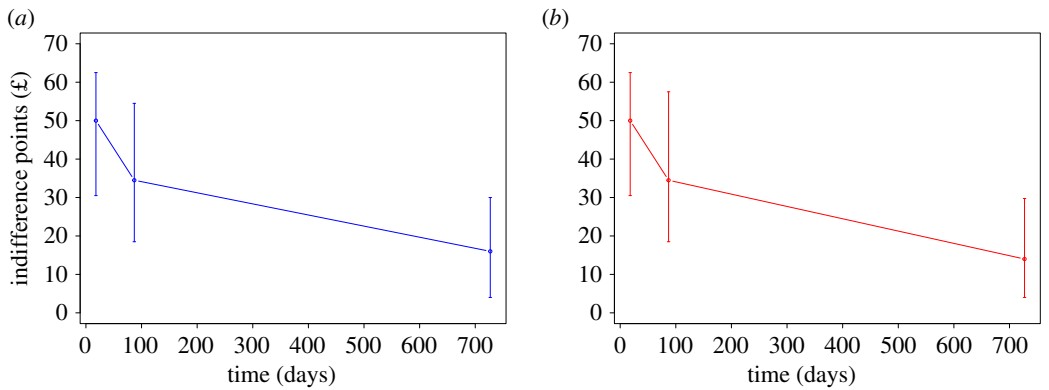

**Figure 1.** Medians of temporal discounting curves of the control (blue) and treated (red) groups. The upper bound of error bars is the third quartile, the lower bound is the first quartile. The distance between them is the interquartile range. Within each plot, the three medians are significantly different (all $p < 0.001$ for paired Wilcoxon tests). This type of visualization was chosen because indifference points are not normally distributed.

**Table 1.** Descriptive statistics. Ladder: McArthur ladder; AUC, area under the curve; 1: study 1; 2: study 2.

| statistic | Mean 1 | Mean 2 | s.d 1 | s.d. 2 | Min 1 | Min 2 | Max 1 | Max 2 |
|---|---|---|---|---|---|---|---|---|
| age (years) | 39.8 | 45.6 | 10.5 | 15.3 | 21.0 | 19.0 | 72.0 | 87.0 |
| education (1–6) | 4.1 | 3.7 | 0.9 | 1.1 | 1.0 | 1.0 | 5.0 | 5.0 |
| ladder choice (1–10) | 5.2 | 4.8 | 1.5 | 1.7 | 1.0 | 1.0 | 9.0 | 10.0 |
| ladder correct | 7.5 | 3.2 | 1.8 | 3.0 | 2.0 | 1.0 | 10.0 | 10.0 |
| total income (£) | 2736 | 1158 | 1258 | 1380 | 1200 | 0 | 12 000 | 10 000 |
| AUC | 20 501 | 20 388 | 12 934 | 12 667 | 709 | 709 | 49 276 | 49 276 |
| total social trust | 0.0 | 0.0 | 2.5 | 2.5 | −5.4 | −5.9 | 5.7 | 5.2 |
| Griskevicius score | 12.7 | 12.0 | 3.3 | 3.6 | 3.0 | 3.0 | 21.0 | 21.0 |

control.value = −0.7 (first quartile) and treat.value = 0.7 (third quartile). The proportion of the total effect that is mediated and all the $p$-values do not vary if other control.value and treat.value are chosen, only $\beta$ values, change. See table 3.

*Causal mediation model.* To test our second main hypothesis (H2) that the effect of the income shock on social trust is mediated by individual discount rates, we used the same steps as for H1 but with the binary variable 'treatment' (0: control, 1: treated) as the treatment variable (i.e. the 'treat' argument in the mediate function). The mediation analysis shows no direct effect of the income shock on social trust and no mediation (average causal mediating effect of temporal discounting: $\beta = 0.001$, $p = 0.85$, proportion of the total effect that is mediated: $-2.5\%$, $p = 0.95$). See table 4.

## 2.5. Discussion

In this study, correcting participants' misperception of their relative income did not have any effect on their temporal discounting and social trust. There are several possible reasons explaining this outcome. First, participants may not have believed the information we provided. We specified that 'the income statistics come from GOV.UK and are based on the whole British population above 18 (taxpayers only)' but this may not have been enough to convince participants that the information was accurate. Given that we did not ask a debriefing question to probe participants' confidence in the income correction, we have no way of knowing whether this was indeed an issue. Second, a correction of people's misperception of their relative income may not be a powerful enough treatment to affect temporal discounting or social trust. In line with this idea, Haushofer *et al.* [47] found that negative income shocks had a larger effect on temporal discounting than positive income shocks.

**Table 2.** Baron & Kenny [54] steps procedure, study 1. X, SES; M, AUC; Y, social trust.

| | dependent variable: | | |
| --- | --- | --- | --- |
| | Y | M | Y |
| | (1) | (2) | (3) |
| X (objective) | 0.168*** | 0.135*** | 0.153*** |
| | (0.034) | (0.034) | (0.034) |
| M | | | 0.111** (0.034) |
| observations | 855 | 855 | 855 |
| $R^2$ | 0.028 | 0.018 | 0.040 |
| adjusted $R^2$ | 0.027 | 0.017 | 0.038 |
| residual s.e. | 0.986 (d.f. = 853) | 0.991 (d.f. = 853) | 0.981 (d.f. = 852) |
| F statistic | 24.767*** (d.f. = 1; 853) | 15.807*** (d.f. = 1; 853) | 17.867*** (d.f. = 2; 852) |
| X (subjective) | 0.264*** (0.033) | 0.197*** (0.034) | 0.248*** (0.034) |
| M | | | 0.083* (0.034) |
| observations | 855 | 855 | 855 |
| $R^2$ | 0.070 | 0.039 | 0.076 |
| adjusted $R^2$ | 0.069 | 0.038 | 0.074 |
| residual s.e. | 0.965 (d.f. = 853) | 0.981 (d.f. = 853) | 0.962 (M = 852) |
| F statistic | 63.821*** (d.f. = 1; 853) | 34.569*** (d.f. = 1; 853) | 35.116*** (d.f. = 2; 852) |

Note: $^*p<0.05$; $^{**}p<0.01$; $^{***}p<0.001$.

**Table 3.** Mediation analysis study 1.

| type of SES | statistic | estimate | 95% CI lower | 95% CI upper |
| --- | --- | --- | --- | --- |
| objective | ACME | 0.016** | 0.005 | 0.03 |
| | ADE | 0.168*** | 0.097 | 0.24 |
| | total effect | 0.185*** | 0.112 | 0.26 |
| | prop. mediated | 0.090** | 0.0276 | 0.19 |
| subjective | ACME | 0.023* | 0.00491 | 0.04 |
| | ADE | 0.347*** | 0.25667 | 0.44 |
| | total effect | 0.369*** | 0.275 | 0.46 |
| | prop. mediated | 0.062* | 0.012 | 0.13 |

Notes: $^*p < 0.05$; $^{**}p < 0.01$; $^{***}p < 0.001$. ACME, average causal mediating effect; ADE, average direct effect.

Karadja *et al.* [32], on the other hand, did find an effect of positive income shocks but in a sample that was mostly negatively biased, unlike our sample, which was mostly positively biased. To sum up, we replicated the known correlations between socio-economic status and temporal discounting on the one hand, and between socio-economic status and social trust on the other hand. More interestingly, we found that temporal discounting and social trust were correlated and that the effect of socio-economic status on social trust was partially mediated by the temporal discounting. However, our study sample was very specific. Specifically, participants were excluded if they overestimated their relative income, which may limit the generalizability of our findings. Therefore, we decided to replicate our main results on a representative sample of the British population.

**Table 4.** Mediation analysis with treatment.

| statistic | estimate | 95% CI lower | 95% CI upper |
|---|---|---|---|
| ACME | 0.001 | −0.017 | 0.02 |
| ADE | −0.060 | −0.191 | 0.07 |
| total effect | −0.058 | −0.189 | 0.08 |
| prop. mediated | −0.025 | −1.210 | 1.16 |

Notes: $^{*}p < 0.05$; $^{**}p < 0.01$; $^{***}p < 0.001$. ACME, average causal mediating effect; ADE, average direct effect.

# 3. Study 2

In this study, our goal was to replicate the mediating effect of temporal discounting on the relationship between socio-economic status and social trust, on a representative sample of the British population. The sample tested in study 1 is indeed not representative because we only included participants who received an income and held wrongly negative beliefs about their relative affluence.

## 3.1. Methods

### 3.1.1. Participants

Participants were recruited online via Prolific Academic (https://www.prolific.co). Our final sample included a representative sample of 1188 participants, stratified across three demographics: age, sex and ethnicity via the representative sample option in Prolific Academic (including 10% more participants for attrition). Sample size was determined *a priori* based on the results of study 1 with the 'Monte Carlo power analysis for indirect effects' application [58].

### 3.1.2. Materials

We used the same materials as in study 1. The questionnaires and discounting task were presented using Qualtrics (https://www.qualtrics.com).

### 3.1.3. Procedure

Participants started with questions about socio-economic status and demographics. Then, they completed a temporal discounting task, and questions on social trust came last. All the measures were the same as those included in study 1.

## 3.2. Data treatment

All data treatment and analyses were carried out in R and were pre-registered (see pre-registration of study 1 https://osf.io/pyj24 and replication pre-registration: https://osf.io/4cjkh). As in study 1, the only deviation from the pre-registered analysis plan is that the exclusion for reaction times was applied at the single-trial level. As in study 1, we focus on the mediation analyses.

 *Exclusion of participants*. The same criteria were applied as those in study 1. This left a sample of 1130 participants with 585 women, 543 men and two others. Ages ranged 19–87, $M = 45.56$, s.d.=15.31.

 *Temporal discounting*. To calculate individual temporal discount rates, we followed the same method as in study 1.

 *Social trust*. As in study 1, the answers obtained to the three questions about social trust were *z*-scored and summed to obtain a single index capturing participants' beliefs on others' level of prosociality.

## 3.3. Statistical analyses

*Correlational mediation model*. Our main hypothesis (H1) is that the relationship between social trust and socio-economic status is mediated by individual discount rates. To test this hypothesis, we conducted regression analyses relying on the causal-steps procedure set forth by Baron & Kenny

**Table 5.** Baron & Kenny [54] steps procedure, study 2. X, SES; M, AUC; Y, social trust.

| | dependent variable: | | |
|---|---|---|---|
| | Y | M | Y |
| | (1) | (2) | (3) |
| X (objective) | 0.136** | 0.096*** | 0.069* |
| | (0.030) | (0.030) | (0.029) |
| M | | | 0.194*** (0.029) |
| observations | 1130 | 1130 | 1130 |
| $R^2$ | 0.019 | 0.009 | 0.046 |
| adjusted $R^2$ | 0.018 | 0.008 | 0.044 |
| residual s.e. | 0.992 (d.f. = 1128) | 0.996 (d.f. = 1128) | 0.978 (d.f. = 1127) |
| F statistic | 21.394** (d.f. = 1; 1128) | 10.383*** (d.f. = 1; 1128) | 27.230*** (d.f. = 2; 1127) |
| X (subjective) | 0.260*** (0.029) | 0.209*** (0.029) | 0.227*** (0.029) |
| M | | | 0.156*** (0.029) |
| observations | 1130 | 1130 | 1130 |
| $R^2$ | 0.067 | 0.043 | 0.091 |
| adjusted $R^2$ | 0.067 | 0.043 | 0.089 |
| residual s.e. | 0.966 | 0.979 | 0.954 |
| F statistic | 81.445*** (d.f. = 1; 1128) | 51.233*** (d.f. = 1; 1128) | 56.197*** (d.f. = 2; 1127) |

Note: $^*p < 0.05$; $^{**}p < 0.01$; $^{***}p < 0.001$.

[54] to detect a mediation effect, following recommendations from Miller *et al.* [55] and Aguinis *et al.* [56]. We tested the significance of the indirect pathway with R mediation package [57] using the function mediate. As in study 1, we set two values (control.value and treat.value) of the treatment to make the contrast.

## 3.4. Results

For descriptive statistics, see table 1.

*Correlational mediation model*. To test our main hypothesis (H1), we conducted regression analyses relying on the causal-steps procedure set forth by Baron & Kenny [54] to detect a mediation effect. The results suggest a partial mediation: the effect size of both objective and subjective socio-economic status is lowered by the addition of temporal discounting in the model (table 5). To test the significance of the mediation pathway, we used R mediation package [57] using the function mediate. We tested again two mediation pathways: one with objective socio-economic status and one with subjective socio-economic status. In both cases, the positive effect of socio-economic status on social trust is partially mediated by temporal discounting (with objective socio-economic status: average causal mediating effect of temporal discounting: $\beta = 0.04$, $p < 0.001$, proportion of the total effect that is mediated: 28%, $p = 0.002$; with subjective socio-economic status: average causal mediating effect of temporal discounting: $\beta = 0.05$, $p < 0.001$, proportion of the total effect that is mediated: 13%, $p < 0.001$). These values are calculated for control.value = −0.80 (first quartile) and treat.value = 0.6 (third quartile) for objective socio-economic status and control.value = −0.73 (first quartile) and treat.value = 0.70 (third quartile) for subjective socio-economic status. The proportion of the total effect that is mediated and all the *p*-values do not vary if other control.value and treat.value are chosen, only $\beta$ values change. See table 6. The mediate function output labels these effects as 'causal mediating effect', but it is important to note that in our case, it is only correlational.

**Table 6.** Mediation analysis with SES, study 2

| type of SES | statistic | estimate | 95% CI lower | 95% CI upper |
|---|---|---|---|---|
| objective | ACME | 0.037*** | 0.020 | 0.06 |
| | ADE | 0.097* | 0.016 | 0.18 |
| | total effect | 0.134** | 0.053 | 0.22 |
| | prop. mediated | 0.277** | 0.128 | 0.71 |
| subjective | ACME | 0.047*** | 0.0270 | 0.07 |
| | ADE | 0.325*** | 0.2352 | 0.41 |
| | total effect | 0.371*** | 0.283 | 0.46 |
| | prop. mediated | 0.126*** | 0.0704 | 0.20 |

Notes: *$p < 0.05$; **$p < 0.01$; ***$p < 0.001$. ACME, average causal mediating effect; ADE, average direct effect.

## 3.5. Discussion

Using a correlational mediation model, we found that the relationship between socio-economic status and social trust is mediated by individual discount rates. This replication on a sample stratified across three demographics: age, sex and ethnicity, confirms the results obtained in study 1.

# 4. General discussion

The two studies reported in this paper investigate the link between socio-economic status and social trust. Our goal was to identify a proximate psychological mechanism to explain this relationship. Our hypothesis was that temporal discounting plays an important role in the association between socio-economic status and social trust. In line with this idea, we found a negative correlation between temporal discounting and social trust in two different samples, including a representative sample of the British population. In addition, we found that temporal discounting partially mediated the effect of socio-economic status on social trust. The mediation scheme brought to light is an important step in the identification of proximate mechanisms explaining the relationship between socio-economic status and behaviours like social trust.

The link between temporal discounting and social trust provides a new understanding of how social trust varies across individuals. Our results are in line with the idea that social trust is modulated by long-term expectations and that it is therefore sensitive to temporal discounting. One possible explanation is that when someone has to decide whether or not to invest in a social interaction, their temporal preferences influence how much they are willing to wait in order to receive a return on their social investment. This is in line with the scientific literature linking temporal discounting and social behaviours, which suggests that several socio-economic gradients in social behaviours can be traced back to variations in temporal preferences [16].

This study and its replication are the first to combine a multi-facet measure of socio-economic status, a precise measure of temporal discounting (that involves more than a single choice), and an explicit measure of social trust. In addition, both our studies are pre-registered, we made sure that we had sufficient power to test our hypotheses and we used a representative sample in the replication study. For those reasons, our results make an important contribution to the limited literature on this topic [22,23].

We also tried to alter temporal discounting using one of its most important predictors, socio-economic status. We borrowed this idea from Karadja *et al.* [32], but there is a growing literature looking at the impact of income and inequality perception on people's psychology. Some studies have focused on inequality perception in a company [59] or in a given society [32,33,60] but political sciences have also investigated this topic [61]. These misperceptions are observed in many societies but are highly variable [60]. Overall, correcting negative misperceptions seems to have less of an effect than correcting positive misperceptions. For instance, people who learn that they are relatively poorer than they thought demand higher levels of redistribution but people finding out that they are richer than they thought do not change their attitudes towards taxation and redistribution [33]. In line with this prior literature, we found no effect of correcting participants' negative misperceptions about their relative income.

Several limitations can also explain this null effect. First, our participants may not have believed us when we told them their real place in the income gradient of the British population because of the way we presented the corrective information. It is possible that other ways of informing participants may be more effective, for example showing them the true income distribution in their society and placing them on it [35,60], or giving them a percentage of people earning less than them [33,35,60]. The possibility that participants did not believe the information shock could also be linked to a negativity bias, which leads people to construe negative information as truer than positive information [62]. In our case, we only tested a positive information shock for ethical reasons but a negative shock might have been easier to believe for participants.

Second, the treatment itself is perhaps not appropriate to impact temporal discounting and social trust. Temporal discounting and social trust are affected by childhood harshness [7,8] and these long-lasting effects may be a lot stronger than our information shock. People may need repeated evidence of their true relative wealth on a longer timescale to express new temporal preferences. Moreover, Fehr [63] and Albanese *et al.* [23] underline the fact that the component of social trust coming from preferences (like temporal preferences) is probably not easily malleable.

Third, we cannot exclude the possibility that our mediation model does not reflect what causally happens in reality and that other mediation models are more appropriate. For instance, alternative models might test whether temporal discounting or social trust lead to differences in socio-economic status, or whether social trust is a mediator of the relationship between socio-economic status and temporal discounting. Given that it is not possible to compare different mediation models statistically, our strategy has been to focus on the strongest model theoretically and to restrain our analyses to that pre-registered model. This model is backed by a number of theoretical assumptions outlined in the introduction. Specifically, Becker & Mulligan [64] have emphasized that wealth causes patience, rather than the opposite. Brandt *et al.* [10] have shown that a model in which income predicts social trust explains the data better than a model with the reversed causal link.

Fourth, the relationship between socio-economic status and social trust is probably mediated by multiple psychological mechanisms beyond temporal discounting. Risk aversion, for instance, plays a role in the risk–benefit calculation that unconsciously guides social trust decisions [23]. Given that social trust is based on reciprocity [15], deciding to trust someone or not indeed requires pitching the certain and immediate costs associated with cooperation against more uncertain and delayed benefits. This risk–benefit balance is affected by two distinct factors: perceived collection risk and perceived waiting cost [65]. A collection risk arises when the benefits of cooperation do not end up materializing because of an unexpected event, such as premature death [65]. Waiting costs refer to the fact that while waiting for a return from a cooperative interaction, an individual could have used her investment for something else with a direct reward [65]. Individuals who have high waiting costs cannot afford to invest resources in cooperative interactions because they need to use these resources in the present. Both waiting costs and collection risks have a strong temporal component and recent studies have indeed found that risk and temporal preferences are somewhat correlated [66]. In addition to temporal and risk preferences, social trust is also affected by people's beliefs about their environment [23]. Social trust therefore also reflects people's trust experience in local interactions [67] and what people have learnt from good and bad past experiences with encounters. These experiences vary according to, for instance, existing values, norms of reciprocity, shared goals within a group [68].

Further research could take several directions. In keeping with our study, it would be interesting to design new treatments that really impact participants' perception of relative affluence, for example, by using more convincing information shocks and checking if participants believe the experimenter. It could also be possible to focus on other methods to shift temporal discounting and to study the consequence on social trust. For instance, several studies have used episodic future thinking to modulate temporal discounting [69–71]. Another possible direction is to conduct longitudinal studies to compute time-series analyses and test whether an exogenous increase in socio-economic status is followed by a decrease of temporal discounting and then an increase in social trust. If an efficient treatment is found to shift people's perception of relative affluence or temporal discounting and if this consequently impacts social trust, the results would be of interest for policy makers when designing interventions to promote prosocial behaviours.

Ethics. The experimental protocol was approved by the local Ethical Committee (Conseil d'Évaluation Éthique pour les recherches en santé-CERES no. 201659) and is in accordance with the Declaration of Helsinki (World Medical Association, 2008). Moreover, before beginning our online experiments, participants had to give their informed consent.

Data accessibility. All data and R scripts are available at https://osf.io/xucgw/.

Competing interests. We declare we have no competing interests.

Funding. This research was supported by the Agence Nationale de la Recherche (EUR FrontCog ANR-17-EURE-0017*).

Acknowledgements. We thank Rita Abdel Sater for her help during the preparation of the pre-registration, especially for the power analysis of study 1. We thank two anonymous reviewers for their insightful comments on our manuscript.

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
