## [Peer Review File · Royal Society Open Science]

Review History

RSOS-200874.R0 (Original submission)

Review form: Reviewer 1

Is the manuscript scientifically sound in its present form?

Yes

Are the interpretations and conclusions justified by the results?

Yes

Is the language acceptable?

Yes

Do you have any ethical concerns with this paper?

No

Have you any concerns about statistical analyses in this paper?

No

Recommendation?

Major revision is needed (please make suggestions in comments)

Comments to the Author(s)

Summary

This manuscript explores the relationship between income and social trust, which are found to be correlated (within and between countries). The hypothesis is that discount rates mediate this relationship, as in: income \rightarrow discount rate \rightarrow social trust. Study 1 was consistent with this. The core finding of the moderation was replicated in Study 2 which was pre-registered.

An experimental manipulation of perceived income (ie by making participants believe they were higher in the income distribution) did not impact social decision making or trust. This runs counter to a longitudinal study [10] showing increases in socioeconomic status (SES) predicted increases in trust, and also to a study showing income shocks alter discount rates [25-27]. Reasons for this discrepancy were discussed.

The manuscript does however provide a clear and balanced introduction into social trust and the factors which influence it. That said, my main expertise is in delay discounting and quantitative methods, less so on social trust.

Overall the manuscript displays high levels of rigour with pre-registered methods, well chosen and processed measures, and analyses. The discount rates were assessed with an online adaptive procedure, deriving from Frye et al. While there were only 3 time points, the adaptive nature of the task means it is likely that the estimation of discount rates are reasonably precise, at least compared to short fixed-questionnaire methods. Use of AUC, and side-stepping the issue of the precise form of the discount functions, is appropriate in this context.

I have only one main point of concern, but I'm reasonably confident that the authors can address it.

Major point

The motivation for the mediation hypothesis (introduced at bottom of page 2) could be made clearer. In particular, temporal discounting is often thought to be associated with risk - that is, one explanation for temporal discounting is that a future reward may not materialise. For example, Sozou (1998) shows that hyperbolic discounting is optimal if we have a certain set of beliefs about the hazard rate (chance a future reward may disappear). There is a line of work which asks whether temporal discounting is in fact caused by risk -- a useful recent paper on this is Johnson et al (2020).

I mention this of course because of the relationship between this concept of risk of a promised future reward disappearing and social trust. If risk were important in determining discount rates, then presumably one would hypothesise that some kind of risk or trust construct would in a causal factor, rather than an effect as conceptualised in this manuscript.

I do not consider this as a fatal flaw in the manuscript. I think that the manuscript could be improved by addressing this issue as one can imagine a number of plausible alternative causal diagrams. In short, more rationale (in the Introduction) or consideration (in the Discussion) should be given for alternative explanations, particularly given that the income shock manipulation did not influence discount rates or social trust.

Minor points

The abstract might want to be updated to reflect the fact that discount rates partially (i.e. not fully) mediates the income / social trust relationship.

I thought that Figure 1 was unnecessary. Citing Myerson, Green, & Warusawitharana (2001) should be sufficient

Figures 2 and 3 are fine given the large numbers of participants. The focus on group level indifference points obscures potentially interesting individual differences - if the authors can think of a way to visualise this given the large numbers of participants then this could be interesting. Alternatively, each figure could be split into subplots, each of which show a turtle or a quartile of participants along the income distribution perhaps. Alternatively, it might be more interesting to see three scatter or density plots of firstly x =income, y =AUC secondly x =AUC, y =social trust, third x =income, y =social trust

References

Sozou, P. D. (1998). On hyperbolic discounting and uncertain hazard rates, 265(1409), 2015–2020.

Johnson, K. L., Bixter, M. T., & Luhmann, C. (2020). Delay discounting and risky choice: Meta-analytic evidence regarding single-process theories. *Judgement and Decision Making*, 15(3), 381–400.

Myerson, J., Green, L., & Warusawitharana, M. (2001). Area under the curve as a measure of discounting. *Journal of the Experimental Analysis of Behavior*, 76(2), 235–243.
<http://doi.org/10.1901/jeab.2001.76-235>

Review form: Reviewer 2

Is the manuscript scientifically sound in its present form?

No

Are the interpretations and conclusions justified by the results?

No

Is the language acceptable?

Yes

Do you have any ethical concerns with this paper?

No

Have you any concerns about statistical analyses in this paper?

No

Recommendation?

Reject

Comments to the Author(s)

The present study investigated a mediation model that higher Socioeconomic status (SES) leads to higher social trust through lower individual discounting rates. Overall, the findings might reflect the overlapping variances of three constructs – as shown in previous studies – rather than revealing the underlying mechanisms of the relation between SES and social trust. In fact, any

alternative model (e.g., TD → SES → ST; SES → ST → TD) will also be able to explain the present data sets. Thus, although the present studies were carefully pre-registered with high enough power to detect expected effects, solid theoretical backgrounds and a specific contribution to the existing field are missing.

Introduction:

- My first struggle was to find a solid theoretical ground for authors' proposed mediation model. The way authors argued that SES relates to social trust through temporal discounting as an individual difference measure seems a bit far-fetched. The only supporting connection between temporal discounting and trust was that trust has some temporal component – people believe that a favor will be returned in the future. In fact, social trust is more closely related to shared goals, intentions, values, and altruism which is shown to dissociate social discounting (and probability discounting) from delayed discounting (e.g., investment in a public-good; Jones & Rachlin, 2009). If any, individual social discounting rate would be a closer construct than the delayed discounting rate.
- How is subjective SES different from objective SES in explaining its relation to temporal discounting and trust? The authors also separately look at the effects of subjective and objective SES in the results but did not provide any separate predictions.
- Page 4 lines 7-8: “experimentally inducing fluctuations in temporal discounting” sounds as if temporal discounting was manipulated. I would suggest to remove “experimentally” and replace “inducing fluctuations” with “assessing fluctuations”

Methods:

- I would also suggest to look at the average value of the two slopes (t1-t2, t2-t3) for the discounting rate because the area under the curve does not consider different levels of steepness for the two time-intervals.
- The authors continuously claimed that the main limitation might be participants not believing in the correction feedback on their relative standing (positive information shock). An alternative design to be able to draw a stronger claim would be to give false feedback (with a manipulation check and thorough debriefing) and compare two groups (positive vs. negative information shock) instead of comparing the positive information group with the no information group.

Results:

- Tables 2 and 3 can be combined (perhaps tag X, M, Y to variables to help readers understand the tables more clearly) – Again, what is the reason for separately assessing objective and subjective SES?

Other tables can be combined across Studies 1 and 2 as well. (e.g., demo, mediation analysis)

- Report how many participants were eventually assigned to the “no information (control)” condition in Study 1.

General discussion:

- The fact that the causal mediation effect was not confirmed and only a partial (not full) mediation occurred in correlational mediations might indicate that the claim for SES relating to social trust through temporal discounting is far-fetched. Authors will need to give convincing explanations for this instead of just focusing on methodological drawbacks.

Decision letter (RSOS-200874.R0)

Dear Mr Guillou

The Editors assigned to your paper RSOS-200874 "Correcting misperceptions of relative income: impact on temporal discounting and social trust." have made a decision based on their reading of the paper and any comments received from reviewers.

Regrettably, in view of the reports received, the manuscript has been rejected in its current form. However, a new manuscript may be submitted which takes into consideration these comments.

We invite you to respond to the comments supplied below and prepare a resubmission of your manuscript. Below the referees' and Editors' comments (where applicable) we provide additional requirements. We provide guidance below to help you prepare your revision.

Please note that resubmitting your manuscript does not guarantee eventual acceptance, and we do not generally allow multiple rounds of revision and resubmission, so we urge you to make every effort to fully address all of the comments at this stage. If deemed necessary by the Editors, your manuscript will be sent back to one or more of the original reviewers for assessment. If the original reviewers are not available, we may invite new reviewers.

Please resubmit your revised manuscript and required files (see below) no later than 03-Mar-2021. Note: the ScholarOne system will 'lock' if resubmission is attempted on or after this deadline. If you do not think you will be able to meet this deadline, please contact the editorial office immediately.

Please note article processing charges apply to papers accepted for publication in Royal Society Open Science (<https://royalsocietypublishing.org/rsos/charges>). Charges will also apply to papers transferred to the journal from other Royal Society Publishing journals, as well as papers submitted as part of our collaboration with the Royal Society of Chemistry (<https://royalsocietypublishing.org/rsos/chemistry>). Fee waivers are available but must be requested when you submit your manuscript (<https://royalsocietypublishing.org/rsos/waivers>).

Thank you for submitting your manuscript to Royal Society Open Science and we look forward to receiving your resubmission. If you have any questions at all, please do not hesitate to get in touch.

on behalf of Dr Simone Schnall (Associate Editor) and Essi Viding (Subject Editor)
openscience@royalsociety.org

Associate Editor Comments to Author (Dr Simone Schnall):
Associate Editor: 1

Comments to the Author:

Thank you for submitting your manuscript to Royal Society Open Science. We've since received two referee reports on your submission - both of which have raised some major points that will require your attention.

Please ensure that you properly address all referee comments in a point-by-point rebuttal upon resubmission of your revised manuscript.

Reviewer comments to Author:

Reviewer: 1

Comments to the Author(s)

Summary

This manuscript explores the relationship between income and social trust, which are found to be correlated (within and between countries). The hypothesis is that discount rates mediate this relationship, as in: income \rightarrow discount rate \rightarrow social trust. Study 1 was consistent with this. The core finding of the moderation was replicated in Study 2 which was pre-registered.

An experimental manipulation of perceived income (ie by making participants believe they were higher in the income distribution) did not impact social decision making or trust. This runs counter to a longitudinal study [10] showing increases in socioeconomic status (SES) predicted increases in trust, and also to a study showing income shocks alter discount rates [25-27]. Reasons for this discrepancy were discussed.

The manuscript does however provide a clear and balanced introduction into social trust and the factors which influence it. That said, my main expertise is in delay discounting and quantitative methods, less so on social trust.

Overall the manuscript displays high levels of rigour with pre-registered methods, well chosen and processed measures, and analyses. The discount rates were assessed with an online adaptive procedure, deriving from Frye et al. While there were only 3 time points, the adaptive nature of the task means it is likely that the estimation of discount rates are reasonably precise, at least compared to short fixed-questionnaire methods. Use of AUC, and side-stepping the issue of the precise form of the discount functions, is appropriate in this context.

I have only one main point of concern, but I'm reasonably confident that the authors can address it.

Major point

The motivation for the mediation hypothesis (introduced at bottom of page 2) could be made clearer. In particular, temporal discounting is often thought to be associated with risk - that is, one explanation for temporal discounting is that a future reward may not materialise. For example, Sozou (1998) shows that hyperbolic discounting is optimal if we have a certain set of beliefs about the hazard rate (chance a future reward may disappear). There is a line of work which asks whether temporal discounting is in fact caused by risk -- a useful recent paper on this is Johnson et al (2020).

I mention this of course because of the relationship between this concept of risk of a promised future reward disappearing and social trust. If risk were important in determining discount rates, then presumably one would hypothesise that some kind of risk or trust construct would in a causal factor, rather than an effect as conceptualised in this manuscript.

I do not consider this as a fatal flaw in the manuscript. I think that the manuscript could be improved by addressing this issue as one can imagine a number of plausible alternative causal diagrams. In short, more rationale (in the Introduction) or consideration (in the Discussion) should be given for alternative explanations, particularly given that the income shock manipulation did not influence discount rates or social trust.

Minor points

The abstract might want to be updated to reflect the fact that discount rates partially (i.e. not fully) mediates the income / social trust relationship.

I thought that Figure 1 was unnecessary. Citing Myerson, Green, & Warusawitharana (2001) should be sufficient

Figures 2 and 3 are fine given the large numbers of participants. The focus on group level indifference points obscures potentially interesting individual differences - if the authors can think of a way to visualise this given the large numbers of participants then this could be interesting. Alternatively, each figure could be split into subplots, each of which show a turtle or a quartile of participants along the income distribution perhaps. Alternatively, it might be more interesting to see three scatter or density plots of firstly x =income, y =AUC secondly x =AUC, y =social trust, third x =income, y =social trust

References

Sozou, P. D. (1998). On hyperbolic discounting and uncertain hazard rates, 265(1409), 2015–2020.

Johnson, K. L., Bixter, M. T., & Luhmann, C. (2020). Delay discounting and risky choice: Meta-analytic evidence regarding single-process theories. *Judgement and Decision Making*, 15(3), 381–400.

Myerson, J., Green, L., & Warusawitharana, M. (2001). Area under the curve as a measure of discounting. *Journal of the Experimental Analysis of Behavior*, 76(2), 235–243.
<http://doi.org/10.1901/jeab.2001.76-235>

Reviewer: 2

Comments to the Author(s)

The present study investigated a mediation model that higher Socioeconomic status (SES) leads to higher social trust through lower individual discounting rates. Overall, the findings might reflect the overlapping variances of three constructs - as shown in previous studies - rather than revealing the underlying mechanisms of the relation between SES and social trust. In fact, any alternative model (e.g., TD \rightarrow SES \rightarrow ST; SES \rightarrow ST \rightarrow TD) will also be able to explain the present data sets. Thus, although the present studies were carefully pre-registered with high enough power to detect expected effects, solid theoretical backgrounds and a specific contribution to the existing field are missing.

Introduction:

- My first struggle was to find a solid theoretical ground for authors' proposed mediation model. The way authors argued that SES relates to social trust through temporal discounting as an individual difference measure seems a bit far-fetched. The only supporting connection between temporal discounting and trust was that trust has some temporal component - people believe that a favor will be returned in the future. In fact, social trust is more closely related to shared goals, intentions, values, and altruism which is shown to dissociate social discounting (and

probability discounting) from delayed discounting (e.g., investment in a public-good; Jones & Rachlin, 2009). If any, individual social discounting rate would be a closer construct than the delayed discounting rate.

- How is subjective SES different from objective SES in explaining its relation to temporal discounting and trust? The authors also separately look at the effects of subjective and objective SES in the results but did not provide any separate predictions.

- Page 4 lines 7-8: “experimentally inducing fluctuations in temporal discounting” sounds as if temporal discounting was manipulated. I would suggest to remove “experimentally” and replace “inducing fluctuations” with “assessing fluctuations”

Methods:

- I would also suggest to look at the average value of the two slopes (t_1-t_2 , t_2-t_3) for the discounting rate because the area under the curve does not consider different levels of steepness for the two time-intervals.

- The authors continuously claimed that the main limitation might be participants not believing in the correction feedback on their relative standing (positive information shock). An alternative design to be able to draw a stronger claim would be to give false feedback (with a manipulation check and thorough debriefing) and compare two groups (positive vs. negative information shock) instead of comparing the positive information group with the no information group.

Results:

- Tables 2 and 3 can be combined (perhaps tag X, M, Y to variables to help readers understand the tables more clearly) – Again, what is the reason for separately assessing objective and subjective SES?

Other tables can be combined across Studies 1 and 2 as well. (e.g., demo, mediation analysis)

- Report how many participants were eventually assigned to the “no information (control)” condition in Study 1.

General discussion:

- The fact that the causal mediation effect was not confirmed and only a partial (not full) mediation occurred in correlational mediations might indicate that the claim for SES relating to social trust through temporal discounting is far-fetched. Authors will need to give convincing explanations for this instead of just focusing on methodological drawbacks.

===PREPARING YOUR MANUSCRIPT===

===PREPARING YOUR REVISION IN SCHOLARONE===

- Ensure that your data access statement meets the requirements at <https://royalsociety.org/journals/authors/author-guidelines/#data>. You should ensure that you cite the dataset in your reference list. If you have deposited data etc in the Dryad repository, please include both the 'For publication' link and 'For review' link at this stage.
- If you are requesting an article processing charge waiver, you must select the relevant waiver option (if requesting a discretionary waiver, the form should have been uploaded at Step 3 'File upload' above).
- If you have uploaded ESM files, please ensure you follow the guidance at <https://royalsociety.org/journals/authors/author-guidelines/#supplementary-material> to include a suitable title and informative caption. An example of appropriate titling and captioning may be found at https://figshare.com/articles/Table_S2_from_Is_there_a_trade-off_between_peak_performance_and_performance_breadth_across_temperatures_for_aerobic_scops_in_teleost_fishes_/3843624.

Author's Response to Decision Letter for (RSOS-200874.R0)

See Appendix A.

RSOS-202104.R0

Review form: Reviewer 1

Is the manuscript scientifically sound in its present form?

Yes

Are the interpretations and conclusions justified by the results?

Yes

Is the language acceptable?

Yes

Do you have any ethical concerns with this paper?

No

Have you any concerns about statistical analyses in this paper?

No

Recommendation?

Accept as is

Comments to the Author(s)

I had one main concern in the original manuscript - relating to the motivation and rationale for the mediation model. For example, social trust is clearly associated in some fashion with risk of future rewards not materialising, which would in turn influence discounting. Thinking along these lines, social trust could be considered as a more upstream factor rather than as the end result as it is in the author's conceptualisation.

The authors address this in the General Discussion by adding a thorough consideration. While this issues are not resolved as such, the changes to the General Discussion do an entirely adequate job of addressing my concern here.

Review form: Reviewer 2**Is the manuscript scientifically sound in its present form?**

Yes

Are the interpretations and conclusions justified by the results?

Yes

Is the language acceptable?

Yes

Do you have any ethical concerns with this paper?

No

Have you any concerns about statistical analyses in this paper?

No

Recommendation?

Accept with minor revision (please list in comments)

Comments to the Author(s)

Overall, I feel that the authors have addressed the issues raised in this revision round quite well in the revised manuscript. Some of the minor points remain as I describe them below.

1. On page 3, "Moving beyond correlations, the goal of our paper is also to study the causal structure of this association by experimentally manipulating perceived income, which is a well-known driver of temporal discounting [REFERENCES NEEDED]."
2. On page 5, In the temporal discounting section, specify what varying delays were applied to the procedure here (I understand that the authors used 3 weeks, 3 months, and 2 years based on the information written in the data treatment).
3. For convenient comparison purposes, Figures 1 and 2 can be combined into one figure. However, in case of a too much overlap between the control and treatment groups, the figures could at least be combined as two graphs next to each other.
4. Tables 4 & 6, include what ACME and ADE stand for (spell them out) in the notes below each table.

5. It is a bit confusing to describe the self-report measure of social trust as more “straightforward measure”. I recommend to describe it as a self-report measure or an explicit measure because one might argue that observing actual behavior (trusting someone in a trust game) is more straightforward than one's subjective opinion about their own attitudes and behaviors.

6. Suggestion: the title of the manuscript does not seem to reflect the main finding of the present studies, mainly because correcting misperceptions of relative income did not show any effect. I would opt for something more direct: “Temporal discounting mediates the effect of objective and subjective economic status on social trust”, but of course, up to the authors.

Decision letter (RSOS-202104.R0)

Dear Mr Guillou

On behalf of the Editors, we are pleased to inform you that your Manuscript RSOS-202104 "Correcting misperceptions of relative income: impact on temporal discounting and social trust." has been accepted for publication in Royal Society Open Science subject to minor revision in accordance with the referees' reports. Please find the referees' comments along with any feedback from the Editors below my signature.

Please submit your revised manuscript and required files (see below) no later than 7 days from today's (ie 06-May-2021) date. Note: the ScholarOne system will 'lock' if submission of the revision is attempted 7 or more days after the deadline. If you do not think you will be able to meet this deadline please contact the editorial office immediately.

on behalf of Essi Viding (Subject Editor)
 openscience@royalsociety.org

Reviewer comments to Author:

Reviewer: 1

Comments to the Author(s)

I had one main concern in the original manuscript - relating to the motivation and rationale for the mediation model. For example, social trust is clearly associated in some fashion with risk of future rewards not materialising, which would in turn influence discounting. Thinking along these lines, social trust could be considered as a more upstream factor rather than as the end result as it is in the author's conceptualisation.

The authors address this in the General Discussion by adding a thorough consideration. While this issues are not resolved as such, the changes to the General Discussion do an entirely adequate job of addressing my concern here.

Reviewer: 2

Comments to the Author(s)

Overall, I feel that the authors have addressed the issues raised in this revision round quite well in the revised manuscript. Some of the minor points remain as I describe them below.

1. On page 3, "Moving beyond correlations, the goal of our paper is also to study the causal structure of this association by experimentally manipulating perceived income, which is a well-known driver of temporal discounting [REFERENCES NEEDED]."
2. On page 5, In the temporal discounting section, specify what varying delays were applied to the procedure here (I understand that the authors used 3 weeks, 3 months, and 2 years based on the information written in the data treatment).
3. For convenient comparison purposes, Figures 1 and 2 can be combined into one figure. However, in case of a too much overlap between the control and treatment groups, the figures could at least be combined as two graphs next to each other.
4. Tables 4 & 6, include what ACME and ADE stand for (spell them out) in the notes below each table.
5. It is a bit confusing to describe the self-report measure of social trust as more "straightforward measure". I recommend to describe it as a self-report measure or an explicit measure because one might argue that observing actual behavior (trusting someone in a trust game) is more straightforward than one's subjective opinion about their own attitudes and behaviors.
6. Suggestion: the title of the manuscript does not seem to reflect the main finding of the present studies, mainly because correcting misperceptions of relative income did not show any effect. I would opt for something more direct: "Temporal discounting mediates the effect of objective and subjective economic status on social trust", but of course, up to the authors.

===PREPARING YOUR MANUSCRIPT===

===PREPARING YOUR REVISION IN SCHOLARONE===

- Any electronic supplementary material (ESM).
- If you are requesting a discretionary waiver for the article processing charge, the waiver form must be included at this step.
- If you are providing image files for potential cover images, please upload these at this step, and inform the editorial office you have done so. You must hold the copyright to any image provided.
- A copy of your point-by-point response to referees and Editors. This will expedite the preparation of your proof.

- Ensure that your data access statement meets the requirements at <https://royalsociety.org/journals/authors/author-guidelines/#data>. You should ensure that you cite the dataset in your reference list. If you have deposited data etc in the Dryad repository, please only include the 'For publication' link at this stage. You should remove the 'For review' link.
- If you are requesting an article processing charge waiver, you must select the relevant waiver option (if requesting a discretionary waiver, the form should have been uploaded at Step 3 'File upload' above).
- If you have uploaded ESM files, please ensure you follow the guidance at <https://royalsociety.org/journals/authors/author-guidelines/#supplementary-material> to include a suitable title and informative caption. An example of appropriate titling and captioning may be found at https://figshare.com/articles/Table_S2_from_Is_there_a_trade-off_between_peak_performance_and_performance_breadth_across_temperatures_for_aerobic_scope_in_teleost_fishes_/3843624.

Author's Response to Decision Letter for (RSOS-202104.R0)

See Appendix B.

Decision letter (RSOS-202104.R1)

Dear Mr Guillou,

I am pleased to inform you that your manuscript entitled "Temporal discounting mediates the relationship between socioeconomic status and social trust." is now accepted for publication in Royal Society Open Science.

Please ensure that you send to the editorial office an editable version of your accepted manuscript, and individual files for each figure and table included in your manuscript. You can send these in a zip folder if more convenient. Failure to provide these files may delay the

processing of your proof. You may disregard this request if you have already provided these files to the editorial office.

on behalf of Essi Viding (Subject Editor)
openscience@royalsociety.org

Appendix A

Dear editor,

Thank you for giving us the opportunity to submit a revised draft of our manuscript entitled "Correcting misperceptions of relative income: impact on temporal discounting and social trust" to *Royal Society Open Science*. We appreciate the time and effort that you and the reviewers have dedicated to providing your feedback on our manuscript. We are grateful to the reviewers for their insightful comments on the paper. We have incorporated their suggestions and highlighted the changes within the manuscript.

Here is a point-by-point response to the reviewers' comments and concerns.

Reviewer 1:

This manuscript explores the relationship between income and social trust, which are found to be correlated (within and between countries). The hypothesis is that discount rates mediate this relationship, as in: income → discount rate → social trust. Study 1 was consistent with this. The core finding of the moderation was replicated in Study 2 which was pre-registered.

An experimental manipulation of perceived income (ie by making participants believe they were higher in the income distribution) did not impact social decision making or trust. This runs counter to a longitudinal study [10] showing increases in socioeconomic status (SES) predicted increases in trust, and also to a study showing income shocks alter discount rates [25-27]. Reasons for this discrepancy were discussed.

The manuscript does however provide a clear and balanced introduction into social trust and the factors which influence it. That said, my main expertise is in delay discounting and quantitative methods, less so on social trust.

Overall the manuscript displays high levels of rigour with pre-registered methods, well chosen and processed measures, and analyses. The discount rates were assessed with an online adaptive procedure, deriving from Frye et al. While there were only 3 time points, the adaptive nature of the task means it is likely that the estimation of discount rates are reasonably precise, at least compared to short fixed-questionnaire methods. Use of AUC, and side-stepping the issue of the precise form of the discount functions, is appropriate in this context.

I have only one main point of concern, but I'm reasonably confident that the authors can address it.

- Major point: The motivation for the mediation hypothesis (introduced at bottom of page 2) could be made clearer. In particular, temporal discounting is often thought to be associated with risk - that is, one explanation for temporal discounting is that a future reward may not materialise. For example, Sozou (1998) shows that hyperbolic discounting is optimal if we have a certain set of beliefs about the hazard rate (chance a future reward may disappear). There is a line of work which asks whether temporal discounting is in fact caused by risk -- a useful recent paper on this is Johnson et al (2020). I mention this of course because of the relationship between this concept of risk of a promised future reward disappearing and social trust. If risk were important in determining discount rates, then presumably one would hypothesise that some kind of risk or trust construct would be a causal factor, rather than an effect as conceptualised in this manuscript. I do not consider this as a fatal flaw in the manuscript. I think that the manuscript could be improved by addressing this issue as one can imagine a number of plausible alternative causal diagrams. In short, more rationale (in the Introduction) or consideration (in the Discussion) should be given for*

alternative explanations, particularly given that the income shock manipulation did not influence discount rates or social trust.

Answer: We agree with the reviewer and acknowledge that the rationale was not sufficiently clear and should have been more explicit. We added several paragraphs in the general discussion to consider alternative explanations and address the topic of risk perception.

« Third, we cannot exclude the possibility that our mediation model does not reflect what causally happens in reality and that other mediation models are more appropriate. For instance, alternative models might test whether temporal discounting or social trust lead to differences in socio-economic status, or whether social trust is a mediator of the relationship between socioeconomic status and temporal discounting. Given that it is not possible to compare different mediation models statistically, our strategy has been to focus on the strongest model theoretically and to restrain our analyses to that pre-registered model. This model is backed by a number of theoretical assumptions outlined in the introduction. Specifically, Becker Mulligan [61] have emphasized that wealth causes patience, rather than the opposite. Brandt et al. [10] have shown that a model in which income predicts social trust explain the data better than a model with the reversed causal link.

Fourth, the relationship between socio-economic status and social trust is likely mediated by multiple psychological mechanisms beyond temporal discounting. Risk aversion, for instance, plays a role in the risk-benefit calculus that unconsciously guide social trust decisions [23]. Given that social trust is based on reciprocity [15], deciding to trust someone or not indeed requires pitching the certain and immediate costs associated with cooperation against more uncertain and delayed benefits. This risk-benefit balance is affected by two distinct factors: perceived collection risk and perceived waiting cost [62]. A collection risk arises when the benefits of cooperation do not end up materializing because of an unexpected event, such as premature death [62]. Waiting costs refer to the fact that while waiting for a return from a cooperative interaction, an individual could have used her investment for something else with a direct reward [62]. Individuals who have high waiting costs cannot afford to invest resources in cooperative interactions because they need to use these resources in the present. Both waiting costs and collection risks have a strong temporal component and recent studies have indeed found that risk and temporal preferences are somewhat correlated [63]. »

10. Brandt MJ, Wetherell G, Henry PJ. 2015 Changes in Income Predict Change in Social Trust: A Longitudinal Analysis: Socioeconomic Status and Trust. *Political Psychology* 36, 761–768.

15. Krueger F, Meyer-Lindenberg A. 2019 Toward a Model of Interpersonal Trust Drawn from Neuroscience, Psychology, and Economics. *Trends in Neurosciences* 42, 92–101.

23. Albanese G, de Blasio G, Sestito P. 2017 Trust, risk and time preferences: evidence from survey data. *International Review of Economics* 64, 367–388.

61. Becker GS, Mulligan CB. 1997 The Endogenous Determination of Time Preference. *The Quarterly Journal of Economics* 112, 729–758.

62. Mell H, Baumard N, André JB. 2017 Both collection risk and waiting costs give rise to the behavioral constellation of deprivation. *Behavioral and Brain Sciences* 40.

63. Johnson KL, Bixter MT, Luhmann CC. 2020 Delay discounting and risky choice: Meta-analytic evidence regarding single-process theories. *Judgment and Decision Making* p. 20.

- *Minor points: The abstract might want to be updated to reflect the fact that discount rates partially (i.e. not fully) mediates the income / social trust relationship.*

Answer: We added « partially » before « mediates » in the abstract.

« Our results indicate that temporal discounting partially mediates the effect of income on social trust in a pre-registered online study on British participants (N = 855). »

« In a second pre-registered study, we replicated the finding that temporal discounting partially mediates the effect of income on social trust in a representative sample of the British population (N = 1130). »

- *I thought that Figure 1 was unnecessary. Citing Myerson, Green, & Warusawitharana (2001) should be sufficient. Myerson, J., Green, L., & Warusawitharana, M. (2001). Area under the curve as a measure of discounting. Journal of the Experimental Analysis of Behavior, 76(2), 235–243. <http://doi.org/10.1901/jeab.2001.76-235>*

Answer: We deleted Figure 1 and cited the suggested article.

« To calculate individual temporal discount rates, we isolated the indifference point for each delay and for each participant and then calculated the area under the discounting curve linking these points [50]. »

50. Myerson J, Green L, Warusawitharana M. 2001 Area Under the Curve as a measure of discounting. Journal of the Experimental Analysis of Behavior 76, 235–243.

- *Figures 2 and 3 are fine given the large numbers of participants. The focus on group level indifference points obscures potentially interesting individual differences - if the authors can think of a way to visualise this given the large numbers of participants then this could be interesting. Alternatively, each figure could be split into subplots, each of which show a turtle or a quartile of participants along the income distribution perhaps. Alternatively, it might be more interesting to see three scatter or density plots of firstly $x=\text{income}$, $y=\text{AUC}$ secondly $x=\text{AUC}$, $y=\text{social trust}$, third $x=\text{income}$, $y=\text{social trust}$*

Answer: We tested the several options using different R packages but given the large number of participants, it was indeed difficult to represent individual curves without producing an unreadable image. Nevertheless, we added two plots in the supplementary materials representing the median and 95 % CI for the quartiles of income distribution.

Reviewer 2:

- *The present study investigated a mediation model that higher Socioeconomic status (SES) leads to higher social trust through lower individual discounting rates. Overall, the findings might reflect the overlapping variances of three constructs – as shown in previous studies – rather than revealing the underlying mechanisms of the relation between SES and social trust. In fact, any alternative model (e.g., $TD \rightarrow SES \rightarrow ST$; $SES \rightarrow ST \rightarrow TD$) will also be able to explain the present data sets. Thus, although the present studies were carefully pre-registered with high enough power to detect expected effects, solid theoretical backgrounds and a specific contribution to the existing field are missing.*

Answer: We agree with the reviewer that the theoretical backgrounds were not developed enough in our previous manuscript version. We now detail the theoretical background that explain the rationale behind the hypothesized relationship between socio-economic status and social trust.

« Third, we cannot exclude the possibility that our mediation model does not reflect what causally happens in reality and that other mediation models are more appropriate. For instance, alternative models might test whether temporal discounting or social trust lead to differences in socio-economic status, or whether social trust is a mediator of the relationship between socioeconomic status and temporal discounting. Given that it is not possible to compare different mediation models statistically, our strategy has been to focus on the strongest model theoretically and to restrain our analyses to that pre-registered model. This model is backed by a number of theoretical assumptions outlined in the introduction. Specifically, Becker Mulligan [61] have emphasized that wealth causes patience, rather than the opposite. Brandt et al. [10] have shown that a model in which income predicts social trust explain the data better than a model with the reversed causal link. »

10. Brandt MJ, Wetherell G, Henry PJ. 2015 Changes in Income Predict Change in Social Trust: A Longitudinal Analysis: Socioeconomic Status and Trust. *Political Psychology* 36, 761–768.

61. Becker GS, Mulligan CB. 1997 The Endogenous Determination of Time Preference. *The Quarterly Journal of Economics* 112, 729–758.

Concerning our contribution to the field, the literature studying a possible link between socio-economic status, temporal discounting and social trust is very scarce (only two studies). Our study intends to fill a gap in our knowledge on this topic, as stated in the manuscript. We also tackled the limitation of using imprecise measures of our three variables of interest (socio-economic status, temporal discounting and social trust).

« This study and its replication are the first to combine a multi-facet measure of socioeconomic status, a precise measure of temporal discounting (that involves more than a single choice), and a straightforward measure of social trust. In addition, both our studies are pre-registered, we made sure that we had sufficient power to test our hypotheses and we used a representative sample in the replication study. For those reasons, our results make an important contribution to the limited literature on this topic [22,23]. »

22. Nguyen Q, Villeval MC, Xu H. 2012 Trust and Trustworthiness Under the Prospect Theory: A Field Experiment in Vietnam. *SSRN Electronic Journal*.

23. Albanese G, de Blasio G, Sestito P. 2017 Trust, risk and time preferences: evidence from survey data. *International Review of Economics* 64, 367–388.

- *Introduction: My first struggle was to find a solid theoretical ground for authors 'proposed mediation model. The way authors argued that SES relates to social trust through temporal discounting as an individual difference measure seems a bit far-fetched. The only supporting connection between temporal discounting and trust was that trust has some temporal component – people believe that a favor will be returned in the future. In fact, social trust is more closely related to shared goals, intentions, values, and altruism which is shown to dissociate social discounting (and probability discounting) from delayed discounting (e.g., investment in a public-good; Jones & Rachlin, 2009). If any, individual social discounting rate would be a closer construct than the delayed discounting rate.*

Answer: We acknowledge that the rationale was not sufficiently clear and should have been more explicit. Social trust is indeed closely related to shared goals, intentions, values, and altruism. However, we see no incompatibility between this idea and the hypothesis that social trust can also be influenced by individual time preferences. The goal of our study was to focus on this temporal component. In the introduction we added theoretical justification and literature on the temporal character of social trust, justifying our focus on temporal

preferences. In the discussion we discuss the possibilities of alternative models and address the question of the different components of social trust: beliefs and preferences.

« While the association between social trust and socioeconomic status is robustly established, the psychological mechanism by which this takes place is not known. An interesting characteristic of social trust is its temporal component, underlined by researchers from several fields, such as education [13], sociology [14] or psychology and neuroscience [15], who agree on the future-oriented character of social trust. For instance, Krueger and Meyer-Lindeberg [15] define social trust as accepting one's vulnerability towards one's expectation regarding "the behavior of another party that will produce some positive outcome in the future". Social trust is based on reciprocity [15] and, thus is associated with certain and immediate costs and with uncertain and delayed benefits. »

13. Verducci S, Schröer A. 2010 pp. 1453–1458. In *Social Trust*, pp. 1453–1458. New York, NY: Springer US.

14. Sztompka P. 1999 *Trust: A sociological theory*. Cambridge University Press.

15. Krueger F, Meyer-Lindenberg A. 2019 Toward a Model of Interpersonal Trust Drawn from Neuroscience, Psychology, and Economics. *Trends in Neurosciences* 42, 92–101.

« In addition to temporal and risk preferences, social trust is also affected by people's beliefs about their environment [23]. Social trust therefore also reflects people's trust experience in local interactions [64] and what people have learnt from good and bad past experiences with encounters. These experiences vary according to, for instance, existing values, norms of reciprocity, shared goals within a group [65]. »

23. Albanese G, de Blasio G, Sestito P. 2017 Trust, risk and time preferences: evidence from survey data. *International Review of Economics* 64, 367–388.

64. Glanville JL, Paxton P. 2007 How do We Learn to Trust? A Confirmatory Tetrad Analysis of the Sources of Generalized Trust. *Social Psychology Quarterly* 70, 230–242.

65. Beugelsdijk S, Klasing MJ. 2016 Diversity and trust: The role of shared values. *Journal of Comparative Economics* 44, 522–540.

- *How is subjective SES different from objective SES in explaining its relation to temporal discounting and trust? The authors also separately look at the effects of subjective and objective SES in the results but did not provide any separate predictions.*

Answer: No separate predictions were made concerning the effects of subjective SES and objective SES on temporal discounting and trust because we had no definitive theory-driven hypotheses about it. The use of a subjective measure was exploratory, as there are few research works examining the association between temporal discounting and subjective SES. We pre-registered separate analyses for these two different measures because we expected that some differences might be observed. Several medical studies have shown that the association between subjective SES and certain health outcomes persist even after adjusting for objective indicators of SES, such as education and income (see Cené et al., 2016; Ghaed & Gallo, 2007). These results indicate that people's perception of their own position in the social hierarchy have important health implications beyond the impact of objective SES. Similarly, social trust could be influenced by the perception of one's position in the social hierarchy independently of one's actual income and education level. If this is the case, it could be expected that subjective SES would be an even better predictor of social trust than objective SES. But to our knowledge, quantitative research evidencing an effect of subjective SES on social trust is lacking. As such we had no theoretical ground to make specific predictions, but all this should have been made clearer in the paper.

We also computed a composite score of global SES combining Objective and Subjective SES even if they measure two different things. We performed the mediation analyses and it did not qualitatively change the results. We reported these analyses in the Supplementary Materials. In the main text, we decided to only report our pre-registered analysis plan and to keep these two variables separated.

Cené, C. W., Halladay, J. R., Gizlice, Z., Roedersheimer, K., Hinderliter, A., Cummings, D. M., ... & DeWalt, D. A. (2016). Associations between subjective social status and physical and mental health functioning among patients with hypertension. *Journal of health psychology*, 21(11), 2624-2635.

Ghaed, S. G., & Gallo, L. C. (2007). Subjective social status, objective socioeconomic status, and cardiovascular risk in women. *Health Psychology*, 26(6), 668.

- *Page 4 lines 7-8: “experimentally inducing fluctuations in temporal discounting” sounds as if temporal discounting was manipulated. I would suggest to remove “experimentally” and replace “inducing fluctuations” with “assessing fluctuations”*

Answer: We removed « experimentally » and replaced « inducing fluctuation » with « assessing fluctuations ».

« Moving beyond correlations, the goal of our paper is also to study the causal structure of this association by experimentally manipulating perceived income, which is a well-known driver of temporal discounting. In this context, our second hypothesis is that assessing fluctuations in temporal discounting should impact social trust. »

- *Methods: I would also suggest to look at the average value of the two slopes (t1-t2, t2-t3) for the discounting rate because the area under the curve does not consider different levels of steepness for the two time-intervals.*

Answer: We pre-registered the use of AUC following recommendations in the temporal discounting literature. The literature reports two different ways of computing a temporal discounting score for individuals. The first one is the use of theoretical discounting functions. These functions can be fitted when enough indifference points are measured per participant. In our case, we only have three points per participant which is not enough to fit these functions. The second one is the use of Area Under the Curve, recommended for example by Myerson et al. (2001). The study of slopes between each couple of delays was not suggested by the literature. Moreover we do not have separate predictions for the first (t1-t2) and the second (t2-t3) slope. Moreover, a slope between two indifference points would be less informative of a participant's temporal discounting than the area under the curve that integrates the curve drawn by three indifference points. To conclude, we think that the use of AUC is justified by the literature and our study design and we decide to stick to the preregistration.

Myerson, J., Green, L., & Warusawitharana, M. (2001). Area under the curve as a measure of discounting. *Journal of the Experimental Analysis of Behavior*, 76(2), 235–243.

- *The authors continuously claimed that the main limitation might be participants not believing in the correction feedback on their relative standing (positive information shock). An alternative design to be able to draw a stronger claim would be to give false feedback*

(with a manipulation check and thorough debriefing) and compare two groups (positive vs. negative information shock) instead of comparing the positive information group with the no information group.

Answer: Participants' deception is generally not considered acceptable in the field of experimental economics (Cooper 2014, Ortmann 2002). First, there are ethical concerns not to use deception (Fisher, 2016). Second, participants pools are often seen as public goods for researchers, especially in specific platforms such as Prolific academic. Given that deception can lead to spoilage of the participant pool, many researchers in our field consider that it should be avoided (Cooper 2014, Ortmann 2002). Indeed, participants may not believe future experimenters if they have been deceived in the past. In the light of all this, we considered that we should only give true, accurate and unharmed information to our participants.

Cooper, D. J. (2014). A note on deception in economic experiments. *Journal of Wine Economics*, 9(2), 111-114.

Ortmann, A., and Hertwig, R. (2002). The costs of deception: Evidence from psychology. *Experimental Economics*, 5(2), 111–131.

Fisher, C. B. (2016). *Decoding the ethics code: A practical guide for psychologists*. Sage Publications.

- *Results: Tables 2 and 3 can be combined (perhaps tag X, M, Y to variables to help readers understand the tables more clearly) – Again, what is the reason for separately assessing objective and subjective SES?*

Answer: We combined tables 2 and 3 and replaced SES by X, AUC by M and Social Trust by Y.

- *Other tables can be combined across Studies 1 and 2 as well. (e.g., demo, mediation analysis)*

Answer: We combined tables of descriptives statistics and combined tables reporting results for Objective and Subjective SES together.

- *Report how many participants were eventually assigned to the “no information (control)” condition in Study 1.*

Answer: We reported the number of participants assigned to the control group (in the Data Treatment subsection of study 1).

- *General discussion: The fact that the causal mediation effect was not confirmed and only a partial (not full) mediation occurred in correlational mediations might indicate that the claim for SES relating to social trust through temporal discounting is far-fetched. Authors will need to give convincing explanations for this instead of just focusing on methodological drawbacks.*

Answer: We acknowledge that social trust is a complex multidimensional construct that is influenced by several variables. A partial rather than a full mediation was therefore expected. Moreover the lack of treatment causal effect is not enough to conclude that our model is

worst than alternative ones. Nevertheless, our data do not allow us to identify the precise causal pathway between several alternatives. In the revised version of the paper, we improved the theoretical background supporting our mediation model in the introduction. We also updated the general discussion to develop alternative explanations.

« Third, we cannot exclude the possibility that our mediation model does not reflect what causally happens in reality and that other mediation models are more appropriate. For instance, alternative models might test whether temporal discounting or social trust lead to differences in socio-economic status, or whether social trust is a mediator of the relationship between socioeconomic status and temporal discounting. Given that it is not possible to compare different mediation models statistically, our strategy has been to focus on the strongest model theoretically and to restrain our analyses to that pre-registered model. This model is backed by a number of theoretical assumptions outlined in the introduction. Specifically, Becker Mulligan [61] have emphasized that wealth causes patience, rather than the opposite. Brandt et al. [10] have shown that a model in which income predicts social trust explain the data better than a model with the reversed causal link.

Fourth, the relationship between socio-economic status and social trust is likely mediated by multiple psychological mechanisms beyond temporal discounting. Risk aversion, for instance, plays a role in the risk-benefit calculus that unconsciously guide social trust decisions [23]. Given that social trust is based on reciprocity [15], deciding to trust someone or not indeed requires pitching the certain and immediate costs associated with cooperation against more uncertain and delayed benefits. This risk-benefit balance is affected by two distinct factors: perceived collection risk and perceived waiting cost [62]. A collection risk arises when the benefits of cooperation do not end up materializing because of an unexpected event, such as premature death [62]. Waiting costs refer to the fact that while waiting for a return from a cooperative interaction, an individual could have used her investment for something else with a direct reward [62]. Individuals who have high waiting costs cannot afford to invest resources in cooperative interactions because they need to use these resources in the present. Both waiting costs and collection risks have a strong temporal component and recent studies have indeed found that risk and temporal preferences are somewhat correlated [63]. In addition to temporal and risk preferences, social trust is also affected by people's beliefs about their environment [23]. Social trust therefore also reflects people's trust experience in local interactions [64] and what people have learnt from good and bad past experiences with encounters. These experiences vary according to, for instance, existing values, norms of reciprocity, shared goals within a group [65]. »

10. Brandt MJ, Wetherell G, Henry PJ. 2015 Changes in Income Predict Change in Social Trust: A Longitudinal Analysis: Socioeconomic Status and Trust. *Political Psychology* 36, 761–768.

15. Krueger F, Meyer-Lindenberg A. 2019 Toward a Model of Interpersonal Trust Drawn from Neuroscience, Psychology, and Economics. *Trends in Neurosciences* 42, 92–101.

23. Albanese G, de Blasio G, Sestito P. 2017 Trust, risk and time preferences: evidence from survey data. *International Review of Economics* 64, 367–388.

61. Becker GS, Mulligan CB. 1997 The Endogenous Determination of Time Preference. *The Quarterly Journal of Economics* 112, 729–758.

62. Mell H, Baumard N, André JB. 2017 Both collection risk and waiting costs give rise to the behavioral constellation of deprivation. *Behavioral and Brain Sciences* 40.

63. Johnson KL, Bixter MT, Luhmann CC. 2020 Delay discounting and risky choice: Meta-analytic evidence regarding single-process theories. *Judgment and Decision Making* p. 20.

64. Glanville JL, Paxton P. 2007 How do We Learn to Trust? A Confirmatory Tetrad Analysis of the Sources of Generalized Trust. *Social Psychology Quarterly* 70, 230–242.

65. Beugelsdijk S, Klasing MJ. 2016 Diversity and trust: The role of shared values. *Journal of Comparative Economics* 44, 522–540.

We look forward to hearing from you in due time regarding our submission and to respond to any further questions and comments you may have.

Sincerely,

Léonard Guillou, Aurore Grandin and Coralie Chevallier.

Appendix B

Dear editor,

Thank you for accepting our manuscript entitled "Correcting misperceptions of relative income: impact on temporal discounting and social trust" to *Royal Society Open Science* subject to minor revision. We appreciate the time and effort that you and the reviewers have dedicated to providing your feedback on our manuscript. We are grateful to the reviewers for their insightful comments on the paper. We have incorporated their suggestions and highlighted the changes within the manuscript.

Here is a point-by-point response to the reviewers' comments and concerns.

Reviewer 1:

Comments to the Author(s)

I had one main concern in the original manuscript - relating to the motivation and rationale for the mediation model. For example, social trust is clearly associated in some fashion with risk of future rewards not materialising, which would in turn influence discounting. Thinking along these lines, social trust could be considered as a more upstream factor rather than as the end result as it is in the author's conceptualisation.

The authors address this in the General Discussion by adding a thorough consideration. While this issues are not resolved as such, the changes to the General Discussion do an entirely adequate job of addressing my concern here.

We would like to thank the first reviewer for his/her feedback.

Reviewer: 2

Comments to the Author(s)

Overall, I feel that the authors have addressed the issues raised in this revision round quite well in the revised manuscript. Some of the minor points remain as I describe them below.

- 1. On page 3, "Moving beyond correlations, the goal of our paper is also to study the causal structure of this association by experimentally manipulating perceived income, which is a well-known driver of temporal discounting [REFERENCES NEEDED]."*

Answer: We have added references to the paragraph:

« Moving beyond correlations, the goal of our paper is also to study the causal structure of this association by experimentally manipulating perceived income. Income [24–26] and relative deprivation [27,28] are indeed well-known drivers of temporal preferences. Economists typically measure temporal preferences using delay discounting experiments in which participants are asked to choose between a smaller but sooner reward - for example \$2 tomorrow - and a larger later reward - for example \$5 in one week. In these types of experiments, lower socioeconomic status individuals consistently display steeper temporal discounting [24–26]. Less educated adolescents, compared to adolescents with higher education, also show the same pattern [29]. In this context, our second hypothesis is that inducing fluctuations in temporal discounting by manipulating perceived relative income should impact social trust. »

24. Green L, Myerson J, Lichtman D, Rosen S, Fry A. 1996 Temporal discounting in choice between delayed rewards: The role of age and income.. *Psychology and Aging* 11, 79–84. doi:10.1037/0882-7974.11.1.79.
25. Harrison GW, Lau MI, Williams MB. 2002 Estimating Individual Discount Rates in Denmark: A Field Experiment. *American Economic Review* 92, 1606–1617. doi:10.1257/000282802762024674.
26. Reimers S, Maylor EA, Stewart N, Chater N. 2009 Associations between a one-shot delay discounting measure and age, income, education and real-world impulsive behavior. *Personality and Individual Differences* 47, 973–978. doi:10.1016/j.paid.2009.07.026.
27. Callan MJ, Shead NW, Olson JM. 2011 Personal relative deprivation, delay discounting, and gambling.. *Journal of personality and social psychology* 101, 955.
28. Tabri N, Shead NW, Wohl MJ. 2017 Me, Myself, and Money II: Relative deprivation predicts disordered gambling severity via delay discounting, especially among gamblers who have a financially focused self-concept. *Journal of Gambling Studies* 33, 1201–1211.
29. Lee NC, de Groot RHM, Boschloo A, Dekker S, Krabbendam L, Jolles J. 2013 Age and educational track influence adolescent discounting of delayed rewards. *Frontiers in Psychology* 4. doi:10.3389/fpsyg.2013.00993.

2. On page 5, In the temporal discounting section, specify what varying delays were applied to the procedure here (I understand that the authors used 3 weeks, 3 months, and 2 years based on the information written in the data treatment).

Answer: The specification of the delays was already written in this section: « In the first block, participants had the choice between a smaller reward in three days, and a larger reward in three weeks. In the second and third blocks, the later delay was set to three months and two years respectively. ». We kept this text but completed another sentence in this section to specify the three delays more clearly.

« In this task, participants had to complete three blocks of an intertemporal choice task with varying delays (three weeks, three months and two years) and varying amounts. »

3. For convenient comparison purposes, Figures 1 and 2 can be combined into one figure. However, in case of a too much overlap between the control and treatment groups, the figures could at least be combined as two graphs next to each other.

Answer: We agree that combining these two figures would be better to compare the groups. The first two delays (3 weeks and 3 months) have the same means for both groups, therefore the overlap would make the graph unreadable. We therefore combined the figures as two graphs next to each other.

4. Tables 4 & 6, include what ACME and ADE stand for (spell them out) in the notes below each table.

Answer: These acronyms are now spelt out in the notes below tables 3, 4 & 6.

« ACME: Average Causal Mediating Effect; ADE : Average Direct Effect. »

5. It is a bit confusing to describe the self-report measure of social trust as more “straightforward measure”. I recommend to describe it as a self-report measure or an explicit measure because one might argue that observing actual behavior (trusting someone in a trust game) is more straightforward than one's subjective opinion about their own attitudes and behaviors.

Answer: « Straightforward » was replaced by « explicit ».

« This study and its replication are the first to combine a multi-facet measure of socioeconomic status, a precise measure of temporal discounting (that involves more than a single choice), and an explicit measure of social trust. »

6. Suggestion: the title of the manuscript does not seem to reflect the main finding of the present studies, mainly because correcting misperceptions of relative income did not show any effect. I would opt for something more direct: “Temporal discounting mediates the effect of objective and subjective economic status on social trust”, but of course, up to the authors.

Answer: We agree on the fact that our previous title did not really reflect the main finding of the studies. We changed the title in a similar way as the one proposed by the reviewer.

« Temporal discounting mediates the relationship between socioeconomic status and social trust. »

We look forward to hearing from you regarding our submission and to respond to any further questions and comments you may have.

Sincerely,

Léonard Guillou, Aurore Grandin and Coralie Chevallier.